# Butanol Synthesis Routes for Biofuel Production: Trends and Perspectives

**DOI:** 10.3390/ma12030350

**Published:** 2019-01-23

**Authors:** Beata Kolesinska, Justyna Fraczyk, Michal Binczarski, Magdalena Modelska, Joanna Berlowska, Piotr Dziugan, Hubert Antolak, Zbigniew J. Kaminski, Izabela A. Witonska, Dorota Kregiel

**Affiliations:** 1Institute of Organic Chemistry, Faculty of Chemistry, Lodz University of Technology, Zeromskiego 116, 90-924 Lodz, Poland; justyna.fraczyk@p.lodz.pl (J.F.); zbigniew.kaminski@p.lodz.pl (Z.J.K.); 2Institute of General and Ecological Chemistry, Faculty of Chemistry, Lodz University of Technology, Zeromskiego 116, 90-924 Lodz, Poland; michal.binczarski@p.lodz.pl (M.B.); modelska.magdalena89@gmail.com (M.M.); izabela.witonska@p.lodz.pl (I.A.W.); 3Institute of Fermentation Technology and Microbiology, Faculty of Biochemistry and Food Sciences, Lodz University of Technology, Wolczanska 171/173, 90-924 Lodz, Poland; joanna.berlowska@p.lodz.pl (J.B.); piotr.dziugan@p.lodz.pl (P.D.); hubert.antolak@p.lodz.pl (H.A.)

**Keywords:** acetone–butanol–ethanol (ABE) fermentation, biofuel, bioethanol, butanol, *Clostridium* spp.

## Abstract

Butanol has similar characteristics to gasoline, and could provide an alternative oxygenate to ethanol in blended fuels. Butanol can be produced either via the biotechnological route, using microorganisms such as clostridia, or by the chemical route, using petroleum. Recently, interest has grown in the possibility of catalytic coupling of bioethanol into butanol over various heterogenic systems. This reaction has great potential, and could be a step towards overcoming the disadvantages of bioethanol as a sustainable transportation fuel. This paper summarizes the latest research on butanol synthesis for the production of biofuels in different biotechnological and chemical ways; it also compares potentialities and limitations of these strategies.

## 1. Introduction

The energy crisis has created strong demand for the development of alternative sources of energy to traditional fossil fuels. Alongside nuclear, solar and wind power, one such alternative is the production of biofuels from plants, cyanobacteria or eukaryotic microalgae. A particularly promising application of biofuels is as liquid car fuels, which can be blended with or even replace gasoline and diesel. Using bioethanol as a liquid fuel could reduce dependence on fossil fuels and greenhouse gas emissions, as well as decrease the acidification, eutrophication and photochemical smog associated with using gasoline [1]. However, ethanol can be used as an additive only at relatively low concentrations, if we are to maintain the standard parameters of the fuel [2,3,4]. 

Bioethanol can be produced from all types of biomass containing mono-, oligo- and polysaccharides [5,6]. However, the use of simple mono- and disaccharides, such as sucrose, simplifies the process of sugar extraction into water and fermentation to ethanol, and significantly decreases the cost of biosynthesis [7,8,9,10,11,12]. Technologies have been under development since the 1950s that use different sources and methods to produce biodiesel from plants [13,14,15,16,17,18,19,20,21,22,23,24]. 

Agricultural residues do offer an economically viable solution for the production of biofuels. Each year, more than 40 million tons of inedible plant material is produced, much of which is discarded. Agricultural residues offer an economically viable solution for the production of biofuels. Abundant and promising sources of biomass include wheat straw, corn stover, switchgrass, salix, spruce, flax shives, hemp hurds, poplar, alfalfa stems and corn cobs, organic waste materials from the inedible parts of plants, and food processing waste [25,26]. Finding ways to turn this woody biomass into biofuel should now be a priority [27]. 

However, lignocellulosic biomass cannot be converted into fuels directly. Additional physical and chemical processing is required. The standard methods are dilute acid pretreatment and steam explosion pretreatment. Enzymatic hydrolysis is considered a more environmentally friendly option, with great potential for the conversion of lignocellulose to biofuel. However, large amounts of expensive commercial cellulases are required during hydrolysis, which substantially increases the cost of saccharification. Recently, new and more cost-effective processes for breaking down cellulose and hemicellulose into their constituent sugars have been developed. In laboratory-scale production, 70–90% yields of soluble carbohydrates have been obtained from corn stover, hardwood and softwood following treatment with a solvent mixture of biomass-derived γ-valerolactone (GVL), water and dilute acid (0.05 % (*w*/*v*) H_2_SO_4_) [28]. After pretreatment, lignocellulosic biomass can be converted by saccharification into a mixture of simple sugars (mainly pentoses, hexoses and disaccharides), which are more suitable for further transformations. Saccharificated biomass can be converted into fuels along several different pathways. These include bioethanol production [29,30], direct bioconversion into butanol or acetone–butanol–ethanol (ABE) mixtures [31] and additional chemical processes for improving the basic biochemical product. The most promising processes involve the dehydration of bioethanol to hydrocarbons or to butanol and/or other higher molecular homologues [32] and the transformation of biomass into fuels via furane derivatives [33]. Several pathways leading from bioethanol to other combustible products have been developed. 

Due to the disadvantages of ethanol as a universal fuel and component in fuel blends, biobutanol is receiving greater attention [34]. First of all, butanol has a higher calorific value than ethanol (the energy content of butanol is 29.2 MJ/L, that of ethanol is 19.6 MJ/L). It also has better miscibility with gasoline and diesel, lower miscibility with water, a higher octane number (96) and is less volatile [35]. Because of its greater hydrophobicity, butanol can be stored in humid conditions. Moreover, it is noncorrosive and can be used in existing combustion engines, in mixtures with gasoline of up to 30% (*v*/*v*) [36,37]. The properties of butanol closely resemble those of modern gasoline (see Table 1). Butanol also has potential as a starting material for various chemical processes, leading to isoprene, isobutene, butane and others [38].

Currently, most of the butanol produced (at a cost of 7.0–8.4 billion dollars per year) is synthesized through chemical processes based on oxo synthesis, Reppe synthesis or crotonaldehyde hydrogenation. However, for economic reasons the products obtained by these methods cannot currently be considered for use as alternative fuel components. Another promising alternative is based on fermentation, using various bioresources and the Gram-positive bacteria *Clostridium* spp. This route produces a mixture of acetone, butanol and ethanol (acetone–butanol–ethanol fermentation, ABE) [38,39]. However, ABE is limited by the high cost of the fermentation substrates, process inhibition by butanol and by the low concentration of butanol in the product, as well as by the high downstream processing costs [40].

## 2. Butanol Production Routes

### 2.1. Biotechnological Processes for Bio-Based Butanol Production

#### 2.1.1. Clostridia as Butanol Producers

Microbial production of butanol was first reported by Louis Pasteur in 1861. However, it was not until the early twentieth century that Chaim Weizmann used spore-forming, anaerobic bacteria from *Clostridium* genera for the production of butanol on an industrial scale [41]. Numerous studies have since focused on the process of butanol fermentation [42,43,44,45,46]. However, microbial butanol production remained more expensive than petrochemical-based processes up to the 1960s [47]. Interest in using clostridial species for acetone–butanol–ethanol fermentation revived in the 1980s, due to the rising price of petroleum [48,49,50,51,52,53].

*Clostridium* spp. bacteria are able to produce organic compounds, acids, alcohols and other solvents, by fermenting a wide spectrum of carbohydrates [54,55,56,57,58,59]. However, of the saccharolytic and mesophilic species capable of butyrate production, only a few are able to produce butanol with high fermentation yields: *Clostridium acetobutylicum*, *C. aurantibutylicum*, *C. beijerinckii*, and *C. tetanomorphum* [60,61,62,63]. 

Batch butanol fermentation is divided into two different stages, (i) acidogenesis (sugar conversion into organic acids) and (ii) solventogenesis (solvent production) [63]. The bacteria grow exponentially in the first phase of fermentation, producing acids (mostly acetate and butyrate). This leads to a decrease in pH to around 4.5. Towards the end of acidogenesis, the production rate falls as the bacterial cells shift their metabolic activity from acidogenesis to solventogenesis, in response to the low pH [64,65]. The acid products induce the solventogenic enzymes necessary for the second stage [66]. In this phase, acetate and butyrate are consumed as substrates for the biosynthesis of acetone and butanol, and no bacterial growth is observed. In turn, organic acids act as co-substrates for solventogenesis [67,68]. The major product of this second phase is butanol, along with an admixture of acetone and ethanol. The presence of solvents affects the cell membrane fluidity and functions [69,70]. At the end of solventogenesis, the concentrations of butanol and other products reaches a level which stops bacterial metabolism [71].

#### 2.1.2. Metabolism

The metabolic pathway for acidogenesis and solventogenesis starts from sugars that are degraded to pyruvate by the Embden-Meyerhof-Parnas (EMP) pathway [72]. Activity by hexokinase (EC 2.7.1.2), glucose-6-phosphate isomerase (EC 5.3.1.9) and pyruvate kinase (EC 2.7.1.40) has been detected in different strains of saccharolytic *Clostridium* spp. [73]. The utilization of a 1 hexose molecule yields 2 molecules of pyru11vate, with the net formation of 2 molecules each of adenosine triphosphate (ATP) and nicotinamide adenine dinucleotide (NADH) [74,75,76]. Solvent-producing clostridia can also use pentoses, via the phosphogluconate pathway parallel to glycolysis [77,78]. In total, the utilization of 3 pentose molecules to pyruvate yields 5 molecules each of both ATP and NADH [54,55,56,57,58,59,60,61,62,63,64,65,66,67,68,69,70,71,72].

As is clear from this pathway description, pyruvate is a key compound in the metabolism of *Clostridium* spp. [73]. The nature of the fermentation is heavily dependent on the activities of three oxidoreductases: [FeFe] hydrogenase, NADH-ferredoxin reductase and NADPH-ferredoxin reductase. The reduced electron carrier -ferredoxin plays an essential role in the fermentation process. It donates electrons either to hydrogen via hydrogenase or to pyridine nucleotides via the ferredoxin-NAD(P) reductase [79,80,81]. Pyruvate-ferredoxin oxidoreductase generates acetyl-CoA, which can be further converted to acetone, acetate or CO_2_ (oxidized products, or to butanol, ethanol or butyrate) reduced products [82,83].

#### 2.1.3. Carbon Sources

*Clostridium* spp. bacteria are known to have the ability to utilize simple as well as complex sugars, including pentoses and hexoses [84,85,86,87]. To degrade polysaccharides into simple monosaccharides (glucose, xylose and arabinose), clostridia secrete large quantities of extracellular enzymes, including amylase, saccharase, glucosidase, glucoamylase, pullulanase and amylopullulanase. These enzymes enable the production of butanol from other carbon substrates, such as corn, sugar beets, sugar cane, potatoes, tapioca or millet [88,89]. Due to their strong amylase activities, *Clostridium* spp. bacteria are particularly effective at utilizing starchy materials, without requiring their initial hydrolysis. Molasses (cane and soy), whey permeate, cassava and Jerusalem artichokes may also be used as carbon sources for butanol production [89]. Although sucrose and starch are good substrates for butanol fermentation, they are not commonly used because of their high price. Moreover, the use of these crops for butanol synthesis may be controversial, due to competition with demand for food [84].

Several stages are required for the production of biobutanol from polymeric saccharides. In upstream processing, prior to butanol fermentation, renewable biomass consisting of starch or lignocellulosics materials is pretreated. The pretreatment method depends on the type of biomass used [87,88,89]. In one study, around 8 g/L of butanol was produced from soy molasses with the use of the *C. beijerinckii* strain, with a yield of 0.1 g/g of substrate [90]. In another study, which used starch-based packing peanuts for continuous production of butanol, 18.9 g/L of butanol were produced from 80.0 g/L of the substrate within 110 h [91]. According to Madihah [92], as much as 16.0 g/L of butanol can be obtained from gelatinized sago starch, with a yield of 0.24 g/g of glucose. Batch fermentation of cassava starch and cassava chip hydrolysate is reported to have resulted in 16.4–16.9 g/L of butanol, with a yield of 0.26–0.35 g/g of substrate [93]. Liu et al. [94] report that 8.8 g/L of butanol can be produced within 72 h of fermentation using wheat bran hydrolysate containing around 53 g/L of reducing sugars, including glucose, xylose and arabinose. Qureshi et al. [95] used wheat straw hydrolysate as a substrate for butanol production with *C. beijerinckii* and obtained 12.0 g/L of butanol as the final result of bacterial fermentation. Recently, fermentation of rice brans using *C. saccharoperbutylacetonicum* has also been found to produce satisfactory results, with 7.7 g/L of butanol produced at a yield of 0.27 g/g of total sugar [96]. In another study, butanol was produced from pretreated deoiled rice bran with a maximum yield 6.48 g/L [97]. *C. acetobutylicum* cells have been found to be capable of growing in a suspension of 10% (*w*/*v*) extruded fresh domestic waste (DOW), producing 3 g/L butanol [98]. The fermentation yield can be raised to 4.2 g/L by adding cellulases and β-glucosidases. Fresh and dried domestic waste can be treated by extrusion, following which the total sugar content is usually between 28% and 39.3% (*w*/*w*), with glucose constituting 18.4–25.1% of the fresh and dried waste, respectively. Claassen et al. [99] described using steam explosion and enzymatic hydrolysis of domestic waste. 

Algal biomass is another source of raw material for butanol fermentation. One advantage of autotrophic macro-algae is that they consume environmental carbon dioxide in order to grow, thereby helping to reduce global warming. Polysaccharides in algal cell walls account for a large proportion of the total carbon content. Several algal species contain large quantities of carbohydrates, and can provide a good substrate for butanol fermentation. For example, the *C. pasteurianum* strain has been reported to produce approximately 14.0–16.0 g/L of mixed solvent, including butanol, from the halophilic microalgae *Dunaliella* with a mixture of 40 g/L of glycerol [84]. Recently, pretreated algal biomass from *Nannochloropsis* sp., *Arthrospira platensis* and wastewater algae has been used as a substrate for butanol synthesis [100,101]. 

It has been suggested that the best strategy for the economical production of butanol is to use sugars extracted from treated cellulosic biomass as fermentation substrates [102,103]. Agricultural, industrial, forest and wood waste residues are composed of cellulose, hemicellulose and pectin [104]. However, obtaining fermentable sugars from these lignocellulosic materials requires physical, chemical and biological pretreatment, separately or in combination [105]. Nevertheless, agricultural residues seem to be the most promising lignocellulosic substrates for butanol synthesis [95,106,107]. Large amounts of this material, such as rejected whole fruits and vegetables, grains, stalks, cobs and husks, are produced during harvesting and processing, which creates environmental pollution as well as economic costs. Food products, especially fruits, vegetables and grains, are wasted during all stages of the production process, due to their delicate nature and short shelf-lives, after which they are deemed unsuitable for human consumption. 

The lack of coordination among food producers and distributors increases food wastage. According to recent studies, farmers lose between 30% and 40% of the value of their fruit and vegetables before their produce reaches the final consumer [108,109,110]. Rejected fruit and vegetables are mostly used as animal feed or for composting. Voget et al. [111] report the possibility of using apple pomace as a substrate for butanol fermentation. Survase et al. [112] have proposed the use of market-rejected and imperfect vegetables (tomatoes, carrots and cabbage) as supplements for improved butanol production. Supplementation with 5% (*w*/*v*) carrot, cabbage or tomatoes increased the maximum values for total solvent production to 9.96 g/L, 10.49 g/L and 10.65 g/L, respectively, as compared to the control value at 7.43 g/L. In light of the literature concerning butanol production from different kinds of biomass, research efforts should now focus on the use of various forms of waste biomass, as well as on identifying the most efficient bacterial strains for industrial processes.

The discussed examples of ABE fermentation using clostridia and carbon sources are presented in Table 2.

#### 2.1.4. Improvement Strategies

In conventional clostridial fermentation processes, butanol yields of 16–17 g/L can be achieved using the best native bacteria. Therefore, one of the main goals of research has been the development of strains to enhance the butanol yield (Figure 1). Research has focused on: (i) the selection of the bacterial strain, (ii) use of less expensive carbon sources to produce butanol, (iii) metabolic engineering strategies, (iv) the process development and (v) biobutanol recovery techniques.

##### Processes Improvement

Simultaneous hydrolysis of wheat straw towards simple sugar and fermentation to butanol is an attractive alternative to the use of expensive glucose in butanol fermentation processes. Qureshi et al. [113,114,115] performed one- and two-stage fermentation trials to assess the performance of the batch process for butanol production from various polysaccharide materials, using different combinations of pretreatment: (i) fermentation with pretreated wheat straw; (ii) separate hydrolysis and fermentation of wheat straw without removing the sediments; (iii) simultaneous hydrolysis and fermentation of wheat straw without agitation; (iv) simultaneous hydrolysis and fermentation of wheat straw with sugar supplementation, and (v) simultaneous hydrolysis and fermentation of wheat straw with agitation by gas stripping. The highest yield of butanol was obtained using the last combination (v). Various monosaccharides were detected in the hydrolysate, namely glucose, xylose, arabinose, galactose and mannose. However, supplementation with sugar helped to improve the productivity of fed-batch fermentation. 

Aside from product inhibition, other factors can affect butanol fermentation. These include substrate inhibition, salt concentration inhibition, the presence of dead cells, low water activity, the accumulation of macromolecules (polysaccharides), nutrient deficiency and O_2_ diffusion when nutrients are added to the fermenter [116,117,118]. It is worth noting that, whereas whey permeates (lactose) at high concentrations of around 200 g/L, it does not inhibit butanol fermentation, while glucose at concentrations greater than 161 g/L does have an inhibitory effect [119,120]. The major inhibitors of the fermentation of lignocellulosic biomass are compounds present in the lignocellulosic hydrolysates, such as salts, furfural, hydroxymethylfurfural, acetic, ferulic, glucuronic, coumaric acids and various phenolic compounds [121]. Coumaric and ferulic acids have been found to act as inhibitors at a concentration of 0.3 g/L. However, furfural and hydroxymethylfurfural at the same concentration stimulate the growth of bacterial cells, and thereby the process of butanol fermentation [122,123]. Cho et al. [124] investigated the effects of model phenolic compounds in lignocellulosic hydrolysates on butanol production by *C. beijerinckii*. At a concentration of 1 g/L, the model phenolic compounds were found to inhibit cell growth by 64–75%, while the production of butanol was completely inhibited. Therefore, detoxification of lignocellulosic hydrolysates is necessary to maximize butanol production from lignocellulosic hydrolysates [125,126]. 

Inhibitors can be successfully removed by treating the hydrolysate with calcium hydroxide or other hydroxides. Qureshi et al. reported that pretreatment with alkaline peroxide significantly improved the butanol yield from enzymatic hydrolysis of wheat straw using *C. beijerinckii* [116]. The treatment of barley straw hydrolysate with lime also improved butanol production, from around 4.5 g/L with a yield of 0.21 g/g total sugar to 18.0 g/L with a yield of 0.29 g/g total sugar [127]. Similar results have been reported with corn stover and switch grass hydrolysates [107]. The high process costs associated with some feedstocks, product toxicity and low product concentrations are a few of the challenges involved in the production of biobutanol. Identifying the challenges involved in converting lignocellulosic biomass to biobutanol and assessing key process improvements can contribute towards making biobutanol more attractive commercially [120]. 

The efficiency of batch and fed-batch fermentation processes may be affected not only by the presence of numerous fermentation inhibitors, but also by the need for additional and time-consuming steps: sterilization of the bioreactors or re-inoculation. These limitations can be avoided by using continuous fermentation processes. The most common strategies for continuous fermentation are free cell systems, immobilized cell systems and cell recycling in the free cell fermentation process, the cells in the fermentation broth move freely, due to mechanical or air-lift agitation. This keeps the microbial cells and nutrients in suspension and helps promote mass transfer. However, fermentation processes using immobilized cells have important advantages over free-cell continuous fermentation processes. These include enhanced fermentation productivity, the possibility of continuous processing on stable fillings and lower recovery and downstream processing costs. Immobilization eliminates the lag phase and enables efficient continuous operation without repeated inoculation. It also improves cells stability and the catalytic effects of biocatalysts. Immobilization may improve genetic stability and protect cells against shear forces [128]. 

Immobilization has been applied in different bioreactors. Using a fibrous bed bioreactor, Huang et al. [129] achieved a significantly higher butanol yield from corn with a strain of *C. acetobutylicum* than could have been expected using conventional continuous fermentation techniques. Using butyric acid as co-substrate shortened the acid-producing phase (acidogenesis) in the feed stream and increased the duration of the solvent-producing phase (solventogenesis). Napoli et al. used a continuous fermentation process in a packed bed reactor for butanol production [130]. According to Qureshi et al. [131], nutrient limitation should be used to avoid excessive cell growth in packed beds. 

A modified and improved version of free-cell continuous fermentation is the cell recycling and bleeding process. This process has been investigated in a continuous butanol production system with high-density *C. saccharoperbutylacetonicum* [132]. To recycle the cells, a membrane module (for filtration) was implanted into the bioreactor. The removal of excess bacterial cells optimized the dilution rate and facilitated cell bleeding, leading to up to six-fold higher yields of butanol in comparison to conventional continuous fermentation without cell recycling. Further research efforts are necessary in order to scale up continuous systems for the industrial production of butanol [117]. 

Current technologies for the production of biobutanol by fermentation include the purification of biobutanol. Sánchez-Ramírez et al. compared steam stripping distillation using distillation columns to distillation columns with a liquid–liquid extraction column. The results showed the second process to be the more efficient design [133].

An interesting flash fermentation technology was used by Mariano et al. [117] in order to obtain higher productivity from butanol synthesis. This method could also reduce distillation costs. The technology consists of three interconnected units: a fermenter, a cell retention system and a vacuum flash vessel. Using this method, final butanol concentrations of more than 20 g/L were obtained. Extractive fermentation is another promising development, which could reduce the butanol inhibition effect. In this method, butanol is simultaneously produced and selectively removed to keep the concentration of butanol in the fermentation medium low [134]. Other common techniques for the removal of butanol include: liquid–liquid extraction, gas stripping, adsorption, perstraction, reverse osmosis and pervaporation [135]. Liquid–liquid extraction, using extractant containing 20% decanol in oleyl alcohol and *C. acetobutylicum* ATCC 4259 enhances butanol formation under conditions of controlled pH [136]. Alone, decanol is toxic to microorganisms; however, used in a mixture with oleyl alcohol and guerbet alcohol this extractant can increase butanol productivity 2.5-fold in comparison to conventional fermentation processes. The identification of further non-toxic compounds or mixtures of compounds could help improve liquid–liquid extraction of butanol from fermentation broth.

Fermentation integrated with gas stripping is an attractive technology for larger-scale biobutanol synthesis. In a novel two-stage gas stripping process integrated with acetone-butanol-ethanol fermentation, more ABE was produced with higher butanol productivity (0.34 g/L · h). This was a result of reduced butanol inhibition caused by butanol recovery. First-stage gas stripping produced a condensate containing 155.6 g/L butanol, and after phase separation an organic phase was obtained containing 610.8 g/L butanol. Second-stage pervaporation produced a condensate with 441.7 g/L butanol, which after mixing with the organic phase from gas stripping gave 521.3 g/L butanol [137]. 

##### Strain Improvement

The mechanism underlying the response of *C. acetobutylicum* to butanol stress is still poorly understood. According to recent studies by Wang et al. [138], glycolysis by *C. acetobutylicum* may be inhibited under butanol stress, while the TCA cycle is be promoted. The key factors determining the metabolic response of *Clostridium* spp. to butanol stress are thought to be changes in the lipid and fatty acid compositions of bacterial cells, to the intracellular metabolism and to the osmoregulator concentrations. The same authors suggest that *C. acetobutylicum* cells change their levels of long acyl chain saturated fatty acids and branched-chain amino acids to adjust their fluidity and maintain the integrity of their cell membranes under butanol stress. Increased levels of some amino acids (threonine, glycine, alanine, phenylalanine, tyrosine, tryptophan, aspartate and glutamate) could also be responsible for increasing the tolerance of *C. acetobutylicum* to butanol. Increased levels of glycerol have likewise been correlated with osmoregulation and the regulating redox balance. These results point towards the possibility of synthesizing butanologenic strains with higher butanol tolerance.

Liu et al. [139] developed a novel approach known as 1-butanoleglycerol storage to enhance butanol tolerance and prevent productive degeneration in *C. acetobutylicum* during long-term preservation. After 12 months under optimal storage conditions at 37 °C, the cell survival rate in a solution containing 16 g/L butanol mixed with 200 g/L glycerol was 80% and the bacterial cells showed enhanced butanol tolerance of 32 g/L. This was around 2-fold higher than for the wild-type strain. Moreover, the butanol yield was slightly higher compared to the control. These results show that the conditions under which cultures are preserved are very important for enhancing butanol tolerance and preventing loss of productivity.

The use of metabolic engineering has the potential to increase butanol production [55]. Strategies to prevent the destruction of bacterial cells by butanol synthesized via fermentation processes include the genetic engineering of high-butanol producing strains [120]. Lin et al. [140] executed random mutagenesis to metamorphose the deoxyribonucleic acid (DNA) sequence of genes responsible for butanol formation. The mutant strain *C. acetobutylicum* ATCC 824 was developed by serial enrichment of diluted n-butanol. The strain was found to have significantly higher butanol tolerance (121%) than the native strain. Another novel mutant was developed from *C. acetobutylicum*, which was treated with a combination of N-methyl-N’-nitro-N-nitrosoguanidine (MMNG), ethyl methane sulphonate and UV exposure [141]. This strain showed greater potency (20%) in molasses and gave higher butanol yields in comparison to the parent strain. 

Systems-level metabolic engineering of clostridia may lead to the discovery of entirely new biosynthetic pathways for butanol, and to the development of new strains which could overcome the current limitations of butanol fermentation by clostridia (Figure 2) [84,142,143,144,145,146,147,148].

Metabolic engineering first requires an analysis of the metabolic system and of the kinetics of its intracellular enzymatic reactions. The selected organism can then be subjected to genetic or environmental modifications. It is necessary to alter not only the protein content of the organism, but also its enzymatic profile. Identifying and modeling the key enzymatic reactions for butanol ratio in *C. acetobutylicum* is thus an important first step towards the construction of metabolically-engineered production strains [55,146,147,148,149]. In the first decade of the twenty-first century, the genomes of two butanol producing clostridia were sequenced in their entirety: *C. acetobutylicum* ATCC 824 and *C. beijerinckii* NCIMB 8052 [150]. Once the butanol and acetone producing genes had been identified, genetic modifications were attempted to decrease or eliminate the production of acetone production during butanol fermentation. TargeTron technology was used to disrupt the acetoacetate decarboxylase gene (adc), which is responsible for acetone production [149]. As a result, butanol production was increased from 70% to 80% and acetone production was reduced to 0.21 g/L. Sequencing the genomes of more hyper-butanol producing bacteria would open further possibilities for genetic engineering to enhance the process of butanol fermentation [151].

Recombinant DNA technology is an attractive tool for improving solvent production by genetic engineering. This technique was first used on the collection strain *C. acetobutylicum* ATCC 824. However, the modified strain was unable to produce acetone and butanol, probably due to the destruction of solvent producing genes (ctfA, ctfB, adc, aad) after serial sub-culturing [152]. Plasmid pSOLI containing these genes was inserted in bacterial mutants. Unfortunately, the engineered strains were still unable to produce butanol and acetone, due to the destruction of the inserted plasmid. Similar results were reported by Sillers et al. [153], who used clostridia as hosts for butanol-producing genes. Due to the genetic complexity of clostridia and the lack of suitable genetic tools, their efforts were unsuccessful.

Other organisms have been investigated as possible hosts for butanol-producing genes. Butanol-producing genes have been most commonly introduced into *E. coli*, *Pseudomonas putida*, *Bacillus subtilis* and *Saccharomyces cerevisiae*. Maximum butanol production of 20 g/L was obtained for the engineered strain of *E. coli* EB243, in which 33 native genes were deleted and 5 heterologous genes introduced. Strain EB243, which produced butanol with a yield of 34% in batch fermentations, showed great potential for industrial applications [154]. In a study by Inui et al., the genes *thiL*, *hbd*, *crt*, *bcd-etfB-etfA*, *adhE1* and *adhE2* from *C. acetobutylicum* ATCC 824 were introduced into *E. coli*, coding acetyl-CoA acetyltransferase, β-hydroxybutyryl-CoA dehydrogenase, 3-hydroxybutyryl-CoA dehydratase, butyryl-CoA dehydrogenase, butyraldehyde dehydrogenase and butanol dehydrogenase [155]. In another study, a strain of *C. saccharobutylicum* with high hemicellulosic activity was isolated, and its genes were inserted into *E. coli*, encoding crotonase, butyryl-CoA dehydrogenase (*bcd*), electron-transport protein subunits A and B, 3-hydroxybutyryl-CoA dehydrogenase, alcohol dehydrogenase, CoA-transferase, acetoacetate decarboxylase and aldehyde dehydrogenase. Almost all of the genes were also expressed in the host bacteria *Lactobacillus brevis* [156]. Successful expression of bcd genes was also achieved in *S. cerevisiae*, but without a significant improvement in butanol production [157]. In summary, recombinant DNA technology in non-clostridial microorganisms has so far proven incapable of improving yields of butanol over native *Clostridium* spp. Focus should now be concentrated on the further development genetic tools for gene expression in host cells [120]. The application of different nanocatalysts to overcome the challenges of biobutanol production is also receiving more interest [55].

### 2.2. Chemical Processes for Producing Butanol from Bio-Ethanol

The classic approach to producing butanol from bio-ethanol proceeds in three steps. First, the commercial product is thoroughly dehydrated to obtain a 100% anhydrous state. It is then oxidized with acetone in the presence of aluminum isopropanolate (Figure 3) to form acetaldehyde. Highly volatile acetaldehyde is separated by fractional distillation. In the second step, acetaldehyde is condensed in a strongly alkaline environment, yielding crotonaldehyde. In the final stage, crotonaldehyde is hydrogenated to butanol by treatment with isopropanol in the presence of titanium or aluminum isopropanolate. 

This multistage transformation could be simplified substantially by using the borrowed hydrogen procedure, also known as the Gerber reaction [158]. This catalyzed reaction sequence has been found to be highly efficient for the conversion of a broad range of alcohols [159,160,161], but not for ethanol [162]. Dehydrogenation of ethanol substrate has been found to be particularly difficult, while the condensation of acetaldehyde is known to lead to a mixture of higher molecular products and polymeric materials (not shown). Nevertheless, a multistep process, including consecutive oxidation of bioethanol affording acetaldehyde, the base catalyzed aldol condensation of acetaldehyde into crotonaldehyde and finally the hydrogenation of crotonaldehyde with partially hydrogenated intermediate products (2-buten-1-ol and/or butanal) over Sr–P hydroxyapatite catalyst, has been reported as giving 1-butanol with 12% conversion and 67% selectivity [163]. 

There are several reports in the literature describing direct catalytic dimerization leading to conversion of ethanol into 1-butanol [164], using stoichiometric or non-stoichiometric hydroxyapatite [165,166,167,168,169,170], solid bases [170,171], some zeolites [172], coconut shell carbon [173] and supported metals (e.g., Ni, Co, Ru) [174,175,176], metallic (Cu, Fe, Co, Ni, Pd) supported systems modified by oxides, e.g., CeO_2_ [177] and supported palladium systems modified by KOH [178] as catalysts. These technologies involve a two-phase process, which is conducted in a fixed bed reactor with conversion in the range of 10–20% and selectivity of up to 70% only in the case when absolute ethanol is used as starting material. Another promising catalytic process used for the transformation of bioethanol to butanol is performed in liquid phase over solid catalysts in a high-pressure mini reactor. As catalysts in the conversion of ethanol to butanol, Ir, Ru, Rh, Pd, Pt, Au, Ag and Ni supported on alumina were used. Under optimized conditions, 18% ethanol conversion and 60% selectivity to butanol were obtained at 250 °C [179,180,181]. Detailed results are summarized in Table 3. 

Fu et al. [182] explored manganese-catalyzed Guerbet-type transformation of ethanol to butanol. This process proceeded selectively to butanol (79%) with the conversion of 12.6% ethanol in the presence of a well-defined manganese pincer complex at 160 °C. This was the first report showing the usefulness of non-noble-metal-catalysts for upgrading ethanol into higher alcohol in homogeneous phase. Kulkarni et al. [183] confirmed the excellent catalytic properties of manganese pincer complexes of the type [(RPNP)-MnBr(CO)_2_] (R = iPr, Cy, tBu, Ph or Ad) for upgrading ethanol to n-butanol. 

A review by Wu et al. [184] provides a detailed list of heterogeneous and homogenous catalysts used in the processes of valorization of ethanol to butanol, and a summary of the proposed reaction mechanisms for these systems. In particular, catalytic processes running in liquid phase [185] are of great interest to the chemical industry, because of their lower energy requirements and consumption of gas materials. Despite the progress made, however, efficient transformation of bioethanol to butanol in the liquid phase remains a challenge. A particularly stubborn problem is the separation of products from mixtures obtained from industrial installations [186]. 

#### One-Step Continuous Process for the Production of Butanol by Catalytic Conversion of Bioethanol in the Sub-/Supercritical State

Fluids in the sub-/supercritical state have properties advantageous for many reactions. This is due to their higher diffusion, lower viscosity and increased solubility reaction components event of diversified properties in supercritical media. This very special phase for ethanol is achieved when the pressure and temperature are increased to near or above their respective critical points, 240.7 °C and 60.60 atm. [187]. Ghazi Askar and Xu [188] describe a continuous process for the coupling of absolute ethanol (>99%) on a bed of solid catalyst in a pressurized microreactor. The reaction was performed over Ni/Al_2_O_3_ (8–27 wt.% Ni) catalysts and on bifunctional catalysts: (Mn_2_O_3_/Al_2_O_3_ + 27% Ni/Al_2_O_3_) and (Mn_2_O_3_ + 27% Ni/Al_2_O_3_). The rate of conversion and the composition of the products obtained were strongly dependent on the reaction conditions (operating temperature/pressure) and the amount of nickel supported on the catalysts. Generally, higher temperatures/pressures promoted the conversion and production selectivity of butanol. However, if the temperature exceeded 300 °C, then more unwanted gaseous products were formed. The most active of the teste catalysts was 8% Ni/Al_2_O_3_. The highest selectivity (61.7%), yield of 1-butanol (21.6%) and rate of ethanol conversion (35%) were achieved using a continuous flow fixed-bed reactor and 8%Ni/Al_2_O_3_ catalysts with a WHSV of 6.4 h-1 at 250 °C under 173.3 atm.

Using a self-manufactured flow system, Dziugan et al. [189] investigated the activity of 8–20% Ni/Al_2_O_3_ catalysts and a mechanical mixture of 5% Pd-8% Fe/Al_2_O_3_ + 8%Ni/Al_2_O_3_, in a continuous process for the coupling of raw spirit (83% of ethanol), rectified alcohol (96% of ethanol) and absolute alcohol (>99% of ethanol) into butanol. Although most studies on the continuous production of butanol use absolute alcohol as the substrate, raw spirit and rectified alcohol can also be used. The results showed that the amount of nickel used in the catalysts was not the key factor determining the catalytic performance of such systems. Only a slight increase in butanol yield was observed in systems with higher amounts of nickel. However, X-ray powder diffraction (XRD), scanning electron microscopy-energy dispersive X-ray spectroscopy (SEM-EDS) and total organic carbon (TOC) studies performed before and after the reaction step (after 200 h of ethanol coupling) showed greater stability in the system with the highest amount of nickel (20% Ni/Al_2_O_3_).

Other tests have been conducted using a two-zone reactor. In a study by Dziugan et al. [189,190], the first zone of the reactor was filled with 8%Ni/Al_2_O_3_ and the second with 5%Pd–8%Fe/Al_2_O_3_. Metals such as Ni and Pd can act as hydrogen transfer catalysts, promoting both the dehydrogenation of ethanol to aldehydes and hydrogenation of the acetaldehyde condensates to heavier alcohols [191]. Metal oxides supporting Ni and Pd are dehydration catalysts [192]. Separate studies have used bimetallic palladium-iron catalyst, because of its particularly high activity in hydrogen transfer reactions [193,194,195]. The use of a two-zone reactor contributed to improve the H/C ratio in the liquid fuel biocomponent. The reaction is summarized in Figure 4.

The reaction presented in Figure 4 is a multistage process, in which ethanol oxidation to acetaldehyde (B) is followed by the aldol reaction of acetaldehyde. The dehydration of the aldol reaction product then leads to the formation of crotonaldehyde (C), which is finally reduced to butanol. Ethanol undergoes dehydrogenation upon contact with the metal centers of Ni/Al_2_O_3_ catalyst, forming acetaldehyde and hydrogen. Acetaldehyde undergoes the aldol reaction in the presence of alumina to form 3-hydroxybutanal. This is converted into crotonaldehyde by dehydration. Crotonaldehyde is reduced by the hydrogen created in the first step (A) of the reaction, forming butylaldehyde. This is further reduced by hydrogen into butanol. In contrast, over strontium phosphate apatite, represented by the general formula Sr_10−α_M_α_(PO_4_)_6−α_(ZO_4_)_βX2_ (where M = Ba, Ca, Mg, Pb, Cd, Fe, Co, Ni, Cu, Zn, La, H, etc., Z = V, Ac, etc., X = OH, F, Cl, etc.), for example, Sr_10_(PO_4_)_6_(OH)_2_, the mechanism of butanol formation from crotonaldehyde in vapor phase assumes the formation 2-buten-1-ol instead of butanal [196].

## 3. Conclusions

Butanol holds particular promise as a liquid transportation fuel, with advantages over bioethanol as part of a wider portfolio of sustainable energy solutions. The research summarized in this review demonstrates the great potential of different methods and processes for increasing both the efficiency of butanol synthesis by biotechnological and conventional chemical routes and the economic profitability of butanol production on a large scale. Genetic technologies can be adapted to improve the characteristics of different butanol-producing bacteria. One of the most promising chemical solutions is the coupling of bioethanol into butanol in a continuous, one-step process. Taken together, the implementation of these strategies for strain improvement, modification of butanol fermentation, recovery processing and chemical catalysis in successful industrial processes indicates that butanol production is likely to become an economically feasible process in the future. 

## Figures and Tables

**Figure 1 materials-12-00350-f001:**
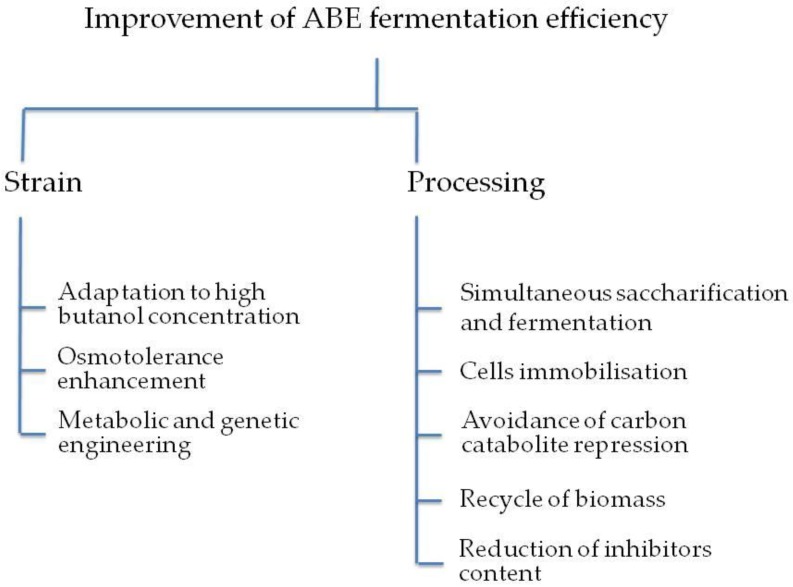
Improvement strategies for biobutanol production.

**Figure 2 materials-12-00350-f002:**
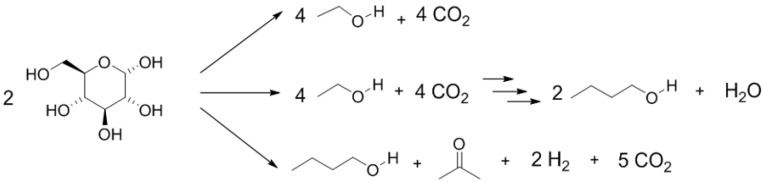
Acetone–butanol–ethanol (ABE) fermentation by clostridia [84,142,143,144,145,146,147,148].

**Figure 3 materials-12-00350-f003:**
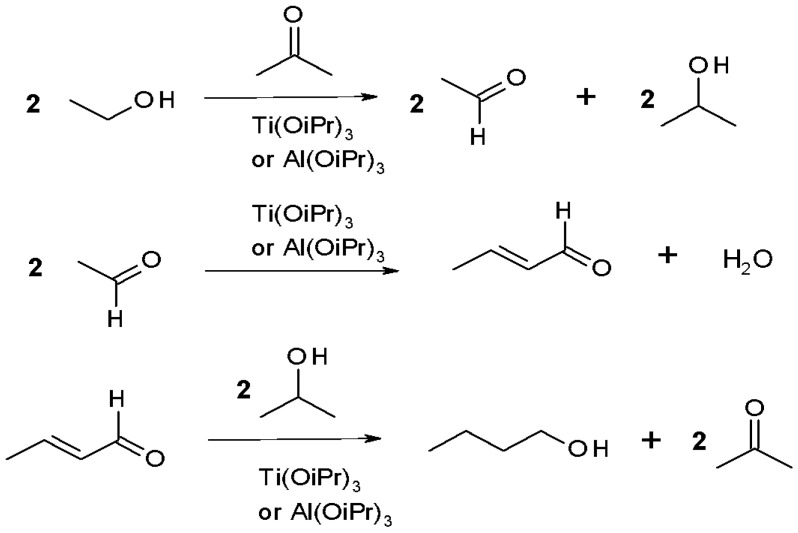
Condensation of dehydrated bioethanol (100% anhydrous state) to butanol in the presence of aluminum (or titanium) isopropanolate.

**Figure 4 materials-12-00350-f004:**
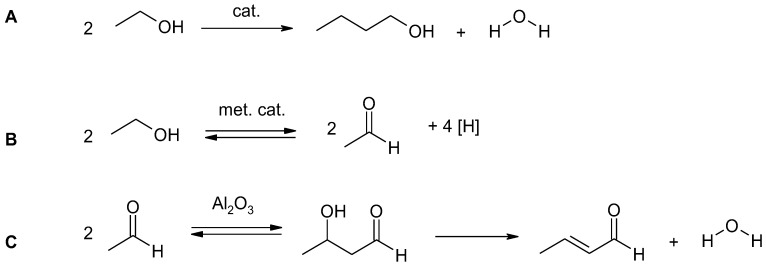
Condensation of bio-ethanol to butanol in the presence of catalysts (**A**) via concomitant elementary transformations (**B**) and (**C**).

**Table 1 materials-12-00350-t001:** Chemical and physical properties of gasoline, diesel, butanol and ethanol [35].

Fuel	Energy Density (MJ/ L)	Air-to-Fuel Ratio	Energy Content/Btu/US gallon	Research Octane Number	Water Solubility (%)
Gasoline	32	14.6	114,000	81-89	Negligible
Diesel	35.5	14.7	130,000	nd	Negligible
Butanol-1	29.2	11.12	105,000	78	7
Ethanol	19.6	8.94	84,000	96	100

nd-no data.

**Table 2 materials-12-00350-t002:** Productivity of butanol by clostridia using different carbon sources.

Strain	Carbon Source	Productivity (g/L)	Reference
*C. beijerinckii*	Soy molasses	8	[90]
*C. beijerinckii* BA101	Starch-based packing peanuts,	18.9	[91]
*C. acetobutylicum*	Gelatinized sago starch	16	[92]
*C. saccharoperbutylacetonicum* N1-4	Cassava starch and cassava chip hydrolysate	16.9	[93]
*C.beijerinckii* ATCC 55025	Wheat bran hydrolysate	8.8	[94]
*C. beijerinckii*	Wheat straw hydrolysate	12	[95]
*C. saccharoperbutylacetonicum*	Rice brans	7.7	[96]
*C. acetobutylicum* YM1	Pretreated deoiled rice bran	6.48	[97]
*C. acetobutylicum*	Fresh domestic wastes	3	[98]
*C. pasteurianum*	Microalgae *Dunaliella* sp. and glycerol	14.0–16.0	[84]
*C. acetobutylicum* DSM 792	Wasted vegetables	9.96–10.65	[112]

**Table 3 materials-12-00350-t003:** One-pot liquid-phase catalytic process for ethanol conversion to 1-butanol over alumina supported with Ru, Rh, Pd, Pt, Au, Ag and Ni catalysts [187].

Catalyst	Manufactured	Product Code	Conver Sion [%]	Selectivity [%]				
				Acet-Aldehyde	Diethyl Ether	Ethyl Acetate	1-Butanol	1,1-Diethoxy Ethane
5%Ru/Al_2_O_3_	Degussa	H213 B/D	2	8	1	30	19	
5%Ru/Al_2_O_3_	La Roche	A 201 (self-prep.)	12	54	3	2	9	31
5%Rh/Al_2_O_3_	Degussa	G214 RA/D	5	4	41	0	35	4
5%Pd/Al_2_O_3_	Degussa	E213 R/D	9	3	64	1	21	2
5%Pt/Al_2_O_3_	Degussa	F 214 XPS/D	3	9	10	9	37	8
6%Ag/Al_2_O_3_	La Roche	A 201 (self-prep.)	2	45	11	6	20	16
20%Ni/Al_2_O_3_	Crossfield	HTC-500	5	5	7	4	62	3
20%Ni/Al_2_O_3_	La Roche	A 201 (self-prep.)	18	43	5	4	37	11
0.8%Au/Al_2_O_3_	Mintek	BC3	6	18	31	15	35	0

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
