# Peer review of "Butanol Synthesis Routes for Biofuel Production: Trends and Perspectives"

_materials, 2019, doi:10.3390/ma12030350_

Round 1

Reviewer 1 Report

The paper deals about the synthesis of butanol both through biological or chemical route. The the topic is interesting, considering the possibility of using butanol as an alternative oxygenate to ethanol.

The paper is well organized and many synthesis routes are described. Anyhow, some minor revisions must be done before the publication and my comments are here reported.

First of all the introduction is too long and it is not suitable for this review on butanol synthesis. Please revise it and focus just on the recent literature and on the topic of the review.

The English must be revised into all the document. Some terms are not appropriate and not correct.

Considering the great amount of data reported, it is necessary to show the main results discussed in some Tables or Figures in order to better understand the butanol productivity and to compare the different synthesis procedures discussed.

Author Response

Journal:Materials (ISSN 1996-1944)

Manuscript ID:materials-417597

Title:Butanol synthesis routes for biofuel production: trends and perspectives

Authors:Beata Kolesinska * , Justyna Fraczyk , Michal Binczarski , Magdalena Modelska , Joanna Berlowska , Piotr Dziugan , Hubert Antolak , Zbigniew J. Kaminski , Izabela A. Witonska , Dorota Kregiel *

Dear Reviewer 1,

We are very grateful for your thorough review of our paper, which has helped us to improve the quality of our publication. We took all your remarks into consideration, and have revised our paper in accordance with your comments:

1.   First of all the introduction is too long and it is not suitable for this review on butanol synthesis. Please revise it and focus just on the recent literature and on the topic of the review.

The Introduction has been significantly shortened and adapted to the title and main content of the publication. The literature was improved with reference to the latest publications in the field.The changes are highlighted in yellow in the text.

2.   The English must be revised into all the document. Some terms are not appropriate and not correct.

The entire text was corrected in terms of language by John Speller (PhD). In addition, we discussed in detail the scientific terms used throughout the publication.

3.   Considering the great amount of data reported, it is necessary to show the main results discussed in some Tables or Figures in order to better understand the butanol productivity and to compare the different synthesis procedures discussed.

As you suggested, instead of presenting the data in the text we added a table and an additional figure in the section concerning the biological synthesis of butanol. We believe this form of data presentation is appropriate and more transparent for the reader.

We hope you will find our article after correction suitable for publication in Materials.

Yours faithfully, 

Dorota Kręgiel & Beata Kolesińska

Reviewer 2 Report

This manuscript describes two pathways for butanol production, either through the biotechnological route, or by the chemical route. The whole paper is lengthy and tedious to read. The authors should carefully check the entire manuscript as the paper is currently not suitable for publication.

Specific comments:

1.     The authors highlighted that chemical route for butanol production has great potential, but described the biotechnological route in more detail in the main text. I wonder what is the focus of this article. If the answer is chemical route for butanol production, the advantage of this route should be added in the manuscript.

2.     Some keywords should be deleted or changed, such as “supercritical conditions” (never appeared in main text), “Fuel biocomponents” (appeared one time in heading) and “catalytic coupling” (appeared one time in abstract).

3.     It is better to remove the sentence “However……parameters of the fuel” in line 33-36 to section 2 as they introduce the same contents.

4.     The part of “3.1. Biotechnological process for bio-based butanol production” is confused and lacks of logicality, and the subtitles were repeated. For example, the subtitles of both 3.1.3 and 3.1.4 are “carbon sources”, and the subtitles of both 3.1.4.1 and 3.1.4.3 (in line 388) are “fermentation processes”. The structure of this part should be reordered and some repeated contents should be deleted to improve readability.

5.     A section of “In situ butanol recovery technology” should be added after the section of “fermentation process”. The contents in line 264-284 could be removed into this section.   "Biotechnology and Bioengineering, 2016, 113(1):120-129." may provide you some suggestions. I suggest the authors adding that paper in the references.

6.     The authors have not made a distinction between butanol titer, butanol yield and butanol ratio. For example, “butanol yields” in line 245 should be “butanol titer”, and “butanol production” in line 400 should be “butanol ratio”. Other similar mistakes should be carefully checked.

7.     Maximum butanol production in E. coli was not 5.8 g/L as strain EB243 (a systematically chromosomally engineered E. coli strain) could produce 20 g/L butanol (Metabolic Engineering, 2017, 44: 284-292.).

8.     The sentence in line 149-150 should be revised to “The major product of this second phase is butanol, with acetone and ethanol as by-products”.

9.     The word “clostridia” should be “clostridia”.

10.  Some minor mistakes:

“butan-1-ol” in Table 1 revised to “Butanol-1”

“negligible” in Table 1 revised to “Negligible”

“0.21 /L” in line 401 revised to “0.21 g/L”

“axcellent” in line 469 revised to “excellent”

“Et all” in line 471 revised to “et al”

11.  There are numerous minor mistakes in the section of “Reference”. The authors need to carefully check that all references have correct and same formats. I did not do a thorough check on this aspect.

Author Response

Journal:Materials (ISSN 1996-1944)

Manuscript ID:materials-417597

Title:Butanol synthesis routes for biofuel production: trends and perspectives

Authors:Beata Kolesinska * , Justyna Fraczyk , Michal Binczarski , Magdalena Modelska , Joanna Berlowska , Piotr Dziugan , Hubert Antolak , Zbigniew J. Kaminski , Izabela A. Witonska , Dorota Kregiel *

Dear Reviewer 2,

We are very grateful for your thorough review of our paper, which has helped us to improve the quality of our publication. We took all your remarks into consideration, and have revised our paper in accordance with your comments:

1.     The authors highlighted that chemical route for butanol production has great potential, but described the biotechnological route in more detail in the main text. I wonder what is the focus of this article. If the answer is chemical route for butanol production, the advantage of this route should be added in the manuscript.

The biotechnological part of the publication and the Introduction have been adjusted in their proportions to reflect the general concept of the article . The changes to the text are highlighted in yellow. 

2.     Some keywords should be deleted or changed, such as “supercritical conditions” (never appeared in main text), “Fuel biocomponents” (appeared one time in heading) and “catalytic coupling” (appeared one time in abstract).

Some keywords have been deleted  and now read as:

Keywords: ABE fermentation; Biofuels; Bioethanol; Butanol; Clostridiumspp.

3.     It is better to remove the sentence “However……parameters of the fuel” in line 33-36 to section 2 as they introduce the same contents.

The Introduction has been shortened significantly and adapted to the title and main content of the publication. The literature has also been improved with reference to the latest publications in the field.The changes to the text are highlighted in yellow.

Regarding notes 4-10:

A considerable fragment of the publication on biological synthesis has been restructured significantly. Some of the results were collected in an additional table and graphically depicted in a diagram. Once again, the nomenclature and terms have been analyzed carefully, and the entire text was proofread by John Speller (PhD), a native speaker. Extensive changes are highlighted in yellow in the text. We are confident that these revised sections will be of interest to readers.

11.  There are numerous minor mistakes in the section of “Reference”. The authors need to carefully check that all references have correct and same formats. I did not do a thorough check on this aspect.

The authors have carefully checked all references. They are now standardized and in the appropriate format for publication in Materials.

We hope that you will find our article after correction suitable for publication in Materials.

Yours faithfully,

Dorota Kręgiel & Beata Kolesińska

Round 2

Reviewer 2 Report

The manuscript is qualified to publish.